# Strong Cellular Immune Response, but Not Humoral, against SARS-CoV-2 in Oncohematological Patients with Autologous Stem Cell Transplantation after Natural Infection

**DOI:** 10.3390/jcm11082137

**Published:** 2022-04-11

**Authors:** Lorena Vigón, Adrián Sánchez-Tornero, Sara Rodríguez-Mora, Javier García-Pérez, Magdalena Corona de Lapuerta, Lucía Pérez-Lamas, Guiomar Casado-Fernández, Gemma Moreno, Montserrat Torres, Elena Mateos, María Aránzazu Murciano-Antón, José Alcamí, Mayte Pérez-Olmeda, Javier López-Jiménez, Valentín García-Gutiérrez, Mayte Coiras

**Affiliations:** 1Immunopathology Unit, National Center of Microbiology, Instituto de Salud Carlos III, 28222 Madrid, Spain; lvhernandez@isciii.es (L.V.); srmora@isciii.es (S.R.-M.); guiomar.casado@externos.isciii.es (G.C.-F.); m.torres@isciii.es (M.T.); emateo@isciii.es (E.M.); 2Hemathology and Hemotherapy Service, Hospital Universitario Ramón y Cajal, 28034 Madrid, Spain; adriansancheztornero@gmail.com (A.S.-T.); madacorona@gmail.com (M.C.d.L.); luciaperezlamas@hotmail.com (L.P.-L.); gmj@richmond.es (G.M.); jljimenez@salud.madrid.org (J.L.-J.); 3AIDS Immunopathology Unit, National Center of Microbiology, Instituto de Salud Carlos III, 28222 Madrid, Spain; eoaz@isciii.es (J.G.-P.); ppalcami@isciii.es (J.A.); 4Family Medicine, Centro de Salud Doctor Pedro Laín Entralgo, 28924 Alcorcón, Spain; aranzazu.murciano@salud.madrid.org; 5Serology Service, Instituto de Salud Carlos III, 28222 Madrid, Spain; mayteperez@isciii.es

**Keywords:** COVID-19, oncohematological disease, autologous transplantation, SARS-CoV-2 neutralizing antibodies, cellular cytotoxicity

## Abstract

Oncohematological patients show a low immune response against SARS-CoV-2, both to natural infection and after vaccination. Most studies are focused on the analysis of the humoral response; therefore, the information available about the cellular immune response is limited. In this study, we analyzed the humoral and cellular immune responses in nine individuals who received chemotherapy for their oncohematological diseases, as well as consolidation with autologous stem cell transplantation (ASCT), after being naturally infected with SARS-CoV-2. All individuals had asymptomatic or mild COVID-19 and were not vaccinated against SARS-CoV-2. These results were compared with matched healthy individuals who also had mild COVID-19. The humoral response against SARS-CoV-2 was not detected in 6 of 9 oncohematological individuals prior to ASCT. The levels of antibodies and their neutralization capacity decreased after ASCT. Conversely, an enhanced cytotoxic activity against SARS-CoV-2-infected cells was observed after chemotherapy plus ASCT, mostly based on high levels of NK, NKT, and CD8+TCRγδ+ cell populations that were able to produce IFNγ and TNFα. These results highlight the importance of performing analyses not only to evaluate the levels of IgGs against SARS-CoV-2, but also to determine the quality of the cellular immune response developed during the immune reconstitution after ASCT.

## 1. Introduction

Oncohematological disorders such as multiple myeloma (MM), Hodgkin lymphoma (HL), or diffuse large B cell lymphoma (DLBCL) may interfere with the efficiency of the immune response against pathogens [1]. The presence of abnormal blood cells, together with recent treatment strategies, impedes appropriate humoral and cellular responses mostly due to the impairment of effector cells [2,3]. Moreover, the immune response developed after vaccination may also be deficient or not be sustained over time due to the therapy received [4].

Autologous stem cell transplantation (ASCT) is a high dose chemotherapy strategy commonly used in oncohematological diseases such as MM or lymphomas to consolidate previous responses [5]. ASCT may profoundly affect the composition and quality of the immune cell populations that retain memory of previous contacts with pathogens and vaccines, and the resetting of the immune system by routine general vaccination of all ASCT receptors is necessary, even in those individuals who had previously passed the infection or had been vaccinated [6]. In fact, current recommendations encourage vaccine administration after immune reconstitution for those pathogens considered to be causative agents of frequent infections after ASCT, such as pneumococcus, influenza virus, or hepatitis B virus (HBV), as well as other infections rarely associated but that present high mortality, such as tetanus, diphtheria, measles, and polio. As most individuals lose their immunity after ASCT, these recommendations apply regardless of the previous serological status [7,8,9]. Therefore, the detection of pre-ASCT immunity against specific pathogens is not a reason to withhold vaccination after transplant.

Due to the emergence of the new coronavirus SARS-CoV-2 in 2019 and the associated disease COVID-19, it is necessary to vaccinate all groups of the population regardless of their immune situation [10]. Accordingly, we need to increase knowledge about the potential effects that the oncohematological disorders and their therapies may have on the development and maintenance of the immune response against COVID-19 induced by natural infection or vaccination with one of the currently approved vaccines.

In this study we have evaluated the humoral and cellular immune responses that persist in oncohematological patients that were subjected to ASCT after having a natural infection with SARS-CoV-2 before and after ASCT. Results were compared with a non-oncologic group who also suffered a previous SARS-CoV-2 infection.

## 2. Materials and Methods

### 2.1. Study Subjects and Samples

Plasma samples were obtained from nine oncohematological patients who had COVID-19 before ASCT (Table 1). Whole blood samples were obtained after ASCT. The inclusion criteria were to be over 18 years old, to suffer an oncohematological disease that needed to be treated with ASCT, and to have a positive SARS-CoV-2 RT-qPCR assay in a nasopharyngeal smear before transplantation. SARS-CoV-2 infection was classified according to the WHO classification [11]. Ten healthy donors who had mild or asymptomatic COVID-19 were recruited as controls. The inclusion criteria for this group were to have a similar age and gender than the individuals with oncohematological disease, to have positive RT-qPCR assay for SARS-CoV-2, and that they only required Primary Healthcare attention during COVID-19.

Blood samples were processed by centrifugation through Ficoll-Hypaque gradient (Pharmacia Corporation, North Peapack, NJ, USA). PBMCs and plasmas were isolated and cryopreserved until analysis. Due to a lack of sample, not all analyses were performed with all samples. Raji cell line (ATCC CCL-86) was provided by the existing collection of Instituto de Salud Carlos III (Madrid, Spain). Vero E6 (African green monkey kidney) cell line (ECACC 85020206) was kindly provided by Dr. Antonio Alcami (CBM Severo Ochoa, Madrid, Spain). Vero E6 and HEK-293T (National Institute for Biological Standards and Control (NIBSC), Potters Bar, UK) cells were cultured in DMEM supplemented with 10% of FCS, 2 mM of L-glutamine and 100 units/mL of penicillin and streptomycin (Lonza, Basel, Switzerland).

### 2.2. Ethical Statement

Individuals with oncohematological disease were recruited from Hospital Universitario Ramón y Cajal (Madrid, Spain). Healthy donors were recruited from Primary Healthcare Center Doctor Pedro Lain Entralgo (Madrid, Spain). All individuals gave informed written consent to participate in the study. Confidentiality and anonymity were ensured by current Spanish and European Data Protection Acts. Protocol for this study (9-4-2021) was performed in accordance with the Helsinki Declaration and it was approved by the Ethics Committees of Hospital Universitario Ramón y Cajal (Favorable report 053-21) and Central Research Commission from the Health Counseling (Comunidad de Madrid, Spain) (Favorable report 20210008).

### 2.3. Phenotyping of B Lymphocytes

Subpopulations of B cells (CD3-CD19+) were analyzed by flow cytometry after the staining of surface markers CD10, CD27, CD20, and CD21 [12]. These cells include immature or transitional cells (CD10+ CD27-), naïve B cells (CD10-CD27-CD21high), tissue-like memory cells (CD10-CD27-CD21low), resting memory cells (CD10-CD27+CD21high), activated memory cells (CD10-CD27+CD21low), and plasmablasts (CD27++CD20-CD21low). Antibodies CD3-PE, CD10-BV421, CD19-BV711, CD20-AlexaFluor700, CD21-FITC, and CD27-PercP-Cy5.5 were purchased from BD Biosciences (San Jose, CA, USA). Data acquisition was performed in BD LSRFortessa X-20 flow cytometer with FACS Diva software (BD Biosciences, San Jose, CA, USA). FlowJo software (Tree Star Inc., Ashland, OR, USA) was used for data analysis.

### 2.4. SARS-CoV-2 Serology

Plasma IgGs against SARS-CoV-2 were analyzed by Euroimmun Anti-SARS-CoV-2 ELISA Assay (Euroimmun, Lübeck, Germany). Semi-quantitative results were obtained by calculating the ratio of extinction of each plasma sample over the calibrator. Results were considered positive when IgG titer was >1.1, values 0.8 to 1.1 were considered undetermined, and values < 0.8 were considered negative. Borderline data were considered positive.

### 2.5. Pseudovirus Neutralization Assays

Pseudotyped SARS-CoV-2 virus pNL4-3Δenv_SARS-CoV-2-SΔ19(G614)_Ren was synthesized as previously described [13,14]. Co-transfection with vector pcDNA-VSV-G was used as the control of specificity. Plasma neutralization activity was measured by pre-incubation of pNL4-3Δenv_SARS-CoV-2-SΔ19(G614)_Ren pseudovirus (10ng p24 Gag per well) with serial dilutions (1/32 to 1/8192) of decomplemented IgG-positive plasma [15]. This mixture was then incubated with Vero E6 cells for 48 h. Cells were then lysed and viral infectivity was assessed by measuring Renilla luciferase activity (Renilla Luciferase Assay, Promega, Madison, WI, USA) with luminometer Centro XS3 LB 960 (Berthold Technologies, Baden-Württemberg, Germany). The titers of neutralizing antibodies were represented as 50% inhibitory dose (ID50) using non-linear regression in GraphPad Prism Software (GraphPad, Inc., San Diego, CA, USA).

### 2.6. Antibody-Dependent Cellular Cytotoxicity Assay

Raji cell line was used as the target to measure ADCC capacity of PBMCs from oncohematological individuals, as previously described [16]. Briefly, Raji cells were previously labeled with PKH67 Green Fluorescent Cell Linker (Merck KGaA, Darmstadt, Germany) and then coated with rituximab (50 g/mL) (Selleckhem, Houston, TX, USA) for 4 h. Labeled Raji cells were co-cultured for 18 h with PBMCs (1:2 ratio) and apoptosis was determined by staining with Annexin V conjugated with phycoerythrin (PE) (Immunostep, Salamanca, Spain). Data were acquired and analyzed in a BD LSRFortessa X-20 flow cytometer (BD Biosciences) using FlowJo_V10 software (TreeStar Inc., Ashland, OR, USA)).

### 2.7. Pseudotyped SARS-CoV-2 Infection for Direct Cellular Cytotoxicity Assay

For analysis of DCC, Vero E6 cells were infected with equal amounts of pNL4-3Δenv_SARS-CoV-2-SΔ19(G614)_Ren [17] and pNL4-3Δenv_SARS-CoV-2 SΔ19(D614)_Ren (100 ng p24 Gag/well) pseudoviruses. After 48 h, Vero cells were co-cultured for 1 h with PBMCs (ratio 1:2). Caspase-3 activity was measured in the monolayer using Caspase-Glo 3/7 Assay system (Promega). Cytotoxic cell populations such as NK, NKT and TCRγδ+ cells were analyzed in the supernatant using specific conjugated antibodies: CD3-PE, CD56-BV605, CD16-PercP, CD8-APC_H7, and TCRγδ-FITC (BD Biosciences). Data were acquired and analyzed in a BD LSRFortessa X-20 flow cytometer (BD Biosciences) using FlowJo_V10 software (TreeStar).

### 2.8. Intracellular Staining of NK, NKT, and CD8+ T Cells

For intracellular staining of IFNγ and TNFα from NK and NKT cells, PBMCs were treated with Hsp70 peptide (Abcam, Cambridge, UK) to stimulate cytolytic activity of NK cells in the presence of brefeldin A (BD Biosciences). Cells were then stained with antibodies against CD3, CD56, and CD16 conjugated to APC, FITC, and PercP, respectively. After fixation and permeabilization with IntraPrep Permeabilization Reagent (Beckman Coulter), cells were stained with antibodies against IFNγ-PE (Beckman Coulter, Indianapolis, IN, USA) or TNFα-PE (Beckman Coulter) and then acquired and analyzed in a BD LSRFortessa X-20 flow cytometer (BD Biosciences) using FlowJo_V10 software (TreeStar).

For intracellular staining of IFNγ and TNFα from CD8+ T cells, PBMCs were treated with PepMix SARS-CoV-2 NCAP (JPT, Berlin, Germany) and CD3/CD49D (BD Biosciences) in the presence of brefeldin A (BD Biosciences). Cells were then stained with antibodies against CD3 and CD8 conjugated to APC and PercP, respectively. After fixation and permeabilization with IntraPrep Permeabilization Reagent (Beckman Coulter), cells were stained with antibodies against IFNγ-PE (Beckman Coulter, Indianapolis, IN, USA) or TNFα-PE (Beckman Coulter) and then acquired and analyzed in a BD LSRFortessa X-20 flow cytometer (BD Biosciences) using FlowJo_V10 software (TreeStar).

### 2.9. Statistical Analysis

Statistical analysis was performed using GraphPad Prism. Statistical differences between two populations were calculated with the Mann–Whitney non-parametric U test. The non-parametric Wilcoxon signed-rank test was used to compare two populations from a set of matched samples pre- and post-treatment. *p* values (*p*) of <0.05 were considered statistically significant.

## 3. Results

### 3.1. Patient Cohorts

An observational, cross-sectional study that included nine patients with an oncohematological disease that required ASCT after being diagnosed with COVID-19 was conducted (Table 1). Six individuals were diagnosed with multiple myeloma (MM), two patients with Hodgkin’s lymphoma (HL), and one patient with diffuse large B cell lymphoma (DLBCL). Most individuals were males (77.8%) and the median age was 64 years (interquartile range (IQR): 57.0 to 70.5). All individuals had mild or asymptomatic COVID-19. Forty-four percent of participants did not receive chemotherapy before COVID-19. After COVID-19 and prior to ASCT, all participants received chemoimmunotherapy. The median time from diagnosis of COVID-19 to ASCT was 5.0 months (IQR 3.9 to 8.8), whereas the median time from ASCT to sampling was 3.4 months (IQR 1.3 to 6.7).

Ten healthy individuals who were diagnosed with mild COVID-19 but did not suffer oncohematological disorders were recruited as controls (Table 1). The median age was 50 years (IQR 29.7 to 64), 40% were male, and the median time from diagnosis of COVID-19 to sampling was 2.8 months (IQR 2.6 to 2.9). The difference in time between diagnosis and sampling should not affect the results in the comparison with participants with oncohematological disease because the immune response against SARS-CoV-2 may remain stable over 7 months after infection [18,19,20].

None of the participants in the study were vaccinated against COVID-19 at the time of sampling.

### 3.2. Changes in B Cells Subpopulations in Individuals after ASCT

As expected, there was a substantial change in the distribution of B cell subpopulations after the ASCT procedure. Plasmablasts and immature B cells were increased 25.0- and 12.0-fold (*p* = 0.0.0056) on average, respectively, in individuals after ASCT, in comparison with healthy donors who passed mild COVID-19. Resting memory and naïve B cell subpopulations were reduced 8.0- (*p* < 0.0001) and 2.1-fold, respectively (Figure 1).

### 3.3. Effect of ASCT on the Levels of IgG against SARS-CoV-2 and Their Neutralization Capacity

Only 3 of 9 individuals with oncohematological disease who had COVID-19 before ASCT developed levels of IgG against SARS-CoV-2 above the threshold of detection (Figure 2A). IgG levels decreased 1.7-fold on average after ASCT (*p* = 0.0039). The neutralizing capacity of these IgGs against SARS-CoV-2 was reduced 4.7-fold after ASCT (Figure 2B). The curves of neutralizing capacity are shown for these three individuals (Figure 2C).

### 3.4. Enhanced ADCC Activity in PBMCs from Oncohematological Individuals after ASCT

Antibody-mediated cytotoxic activity against rituximab-coated Raji cells of peripheral blood lymphocytes (PBMCs) from oncohematological patients was increased 2.1-fold on average after ASCT, in comparison with healthy donors who recovered from mild COVID-19 (Figure 3), but this difference was almost significant (*p* = 0.0592).

### 3.5. Direct Cytotoxic Activity of PBMCs against SARS-CoV-2-Infected Cells

Direct cellular cytotoxicity (DCC) against SARS-CoV-2-infected cells of PBMCs from oncohematological individuals after ASCT was increased 1.5-fold in comparison with healthy donors who recovered from mild COVID-19 (Figure 4A). Cytotoxic populations with Natural Killer (NK) (CD3-CD56+CD16+), Natural Killer T (NKT) (CD3+CD56+CD16+), and CD8+TCRγδ+ T cell phenotypes were increased 1.9- (*p* = 0.0311), 1.9- (*p* = 0.0592), and 1.6-fold, respectively, in comparison with healthy donors (Figure 4B).

### 3.6. Synthesis of Cytokines by Cytotoxic Populations

The ability of the PBMCs from oncohematological patients to release pro-inflammatory cytokines after ASCT was analyzed in a subgroup of individuals due to a lack of sample. The production of interferon gamma (IFNγ) and tumor necrosis factor-alpha (TNFα) by NK cells upon stimulation with Hsp70 was increased, although these results did not achieve statistical significance (Figure 5A). This increased activity of cytotoxic population was also observed in NKT cells, which expression of IFNγ was increased 4.8-fold (*p* = 0.0079), in comparison with healthy individuals who had mild COVID-19 (Figure 5B). The release of TNFα was also increased 12.1-fold (*p* = 0.0317) in CD8+ T cells from oncohematological patients after ASCT upon stimulation with SARS-CoV-2 peptides (Figure 5C). 

## 4. Discussion

Individuals with oncohematological neoplasms present alterations in their immune response due to the hematological disease or the immunosuppressive treatment. For this reason, these patients are most vulnerable to the infectious disease caused by the new coronavirus SARS-CoV-2 (COVID-19) [21,22,23]. Recent data have shown significantly impaired immune responses to SARS-CoV-2 exposure or vaccination in comparison with the general population, but most of these studies have been focused on the humoral immune response, with limited information about the cellular response that is developed against the infected cells [24,25,26,27]. In this study, we analyzed the quality and persistence of the humoral and cellular immune responses developed by nine individuals who were diagnosed with MM, HL, or DLBCL and were naturally infected with SARS-CoV-2 before receiving ASCT. Due to the necessity of an immune reconstitution after ASCT, it was expected that both humoral and cellular responses developed against SARS-CoV-2 prior to ASCT were substantially modified after this procedure, but this remained to be determined.

A poor humoral response after SARS-CoV-2 infection was observed in our cohort, as only 3 of 9 patients developed IgGs against SARS-CoV-2 before ASCT. These results were in accordance with previous reports from oncohematological patients with COVID-19 [28,29,30] and may be related to the blood disorder or to the treatment received at the time of infection. Two of these individuals with detectable IgGs were diagnosed with HL and on treatment with ABVD (adriamycin-bleomycin-vinblastine-dacarbazine), whereas the third participant with MM was on watchful waiting. A humoral response developed against SARS-CoV-2 was greatly diminished after ASCT in all three individuals, and the neutralizing capacity was reduced as well. Along with ASCT, which may negatively influence the humoral response, these individuals were treated with brentuximab vedotin and/or corticoids such as dexamethasone or prednisolone, which may also contribute to the reduction in IgG levels after ASCT. The other participants (6/9) were all treated with VRD (bortezomib-lenalidomide-dexamethasone), except the individual diagnosed with DLBCL who was treated with PBR (polatuzumab vedotin-bendamustine-rituximab), being the last two drugs able to eliminate B cells.

The reconstitution of the immune system after ASCT is variable and highly influenced by the immune characteristics of the recipient, the donor, and the treatment received before and after transplant [31]. Cells involved in innate immunity such as neutrophils, monocytes, and NK cells are usually restored sooner than cells related to the acquired immune response, such as B and T cells. It has been determined that CD8+ T cells reconstitution is faster than CD4+ T cells, which may occur after 3 months or later [32], but there is limited information about the reconstitution of B cell subpopulations. B cell count is usually recovered within one year after transplant [33] but normal function may be compromised during the first two years, due also to delayed CD4 recovery [34]. Consequently, complete immune restoration may take 2 years after ASCT [35]. Moreover, blood disorders are characterized by B-cell development deregulation with increased numbers of plasma cells in the bone marrow, which may interfere with an adequate humoral response against SARS-CoV-2 before ASCT [36]. This situation does not improve after ASCT because B cell immaturity may continue until complete restoration. In fact, the count of immature B cells in our cohort was increased more than 10-fold in comparison with healthy donors, which may account for the decrease in circulating IgGs against SARS-CoV-2 after ASCT due to the proportional loss of memory subpopulations. Interestingly, IgGs did not disappear completely more than 3 months after ASCT in patients_002, _005, and _008, which may be related to the huge level of surviving plasmablasts observed in the peripheral blood of these individuals. This sustained increased levels of plasmablasts have been observed after ASCT [37] and also after infection with SARS-CoV-2 [16].

Therefore, the impaired humoral response observed in our cohort may be related to the delayed CD4+ T cell reconstitution, which has a central role in the efficiently acquired immune response [38]. Surprisingly, a significant increase in ADCC was observed in PBMCs from oncohematological individuals after ASCT. Due to the absence of detectable IgGs against SARS-CoV-2 in these patients, rituximab-coated Raji cells were used as the target. Despite the impaired humoral response, potent unspecific ADCC activity was observed in these patients. Moreover, PBMCs from these individuals also showed high levels of activated, functional NK and NKT cells, as well as highly cytotoxic CD8+ T cells with TCR, which were related to an increased specific DCC against SARS-CoV-2 infected cells.

The recovery of CD4 counts may take several years after ASCT in older patients [39] and our cohort had a median age of 64 years. However, the immune reconstitution of NK cells occurs soon after ASCT and these cells have been related to prolonged remission of neoplasms [38]. NK cells have a dual cytotoxic activity because they can mediate ADCC through the binding of CD16 to the target cell, but also to induce antibody-independent DCC that is not mediated by the major histocompatibility complex (MHC) [40]. Therefore, the cytotoxic activity of PBMCs from oncohematological patients was more likely related to innate immunity mediated by NK and NKT cells than to CD4-mediated acquired immunity developed before ASCT against SARS-CoV-2. On the other hand, six individuals from our cohort were treated with lenalidomide (5/9) or its analogue thalidomide (1/9). Lenalidomide is an immunomodulatory drug that induces an antiviral Th1 CD4-mediated response that stimulates NK cell activity through the release of IFNγ and IL-2 [41,42], increasing NK-mediated ADCC [43]. Therefore, we cannot rule out that the treatment received after ASCT was contributing to this enhanced NK activity.

In conclusion, treatment of oncohematological disorders with ASCT and the subsequent pharmacological treatment greatly affected not only the levels of IgGs specific of SARS-CoV-2 after natural infection but also their neutralizing capacity, which correlated with previous reports [7,9,44,45]. Therefore, similar deficient humoral responses may be expected in oncohematological patients who receive COVID-19 vaccines prior to ASCT. In contrast, a consistent cellular response to SARS-CoV-2 was observed in these patients, including those without a humoral response. This response was related to an enhanced restored NK activity or to the treatment received after ASCT, although we cannot rule out that some memory cells may remain after the mobilization and collection of CD34+ cells for subsequent infusion [46] or that memory cells located in the lymph nodes were not depleted like those in the bone marrow, allowing the generation of a cellular response during the immune reconstitution. The main limitation of this study was the number of patients recruited, but statistically significant results were obtained, nonetheless. Moreover, our cohort is now very rare because all oncohematological patients have already been vaccinated. To the best of our knowledge, this is the first report describing the enhanced cellular immune response developed by individuals with blood disorders previously exposed to SARS-CoV-2 that underwent ASCT. Finally, the maintenance of the immune response developed in oncohematological individuals before ASCT cannot be determined exclusively by analyzing the humoral response, but the cytotoxic activity should also be evaluated during the immune reconstitution.

## Figures and Tables

**Figure 1 jcm-11-02137-f001:**
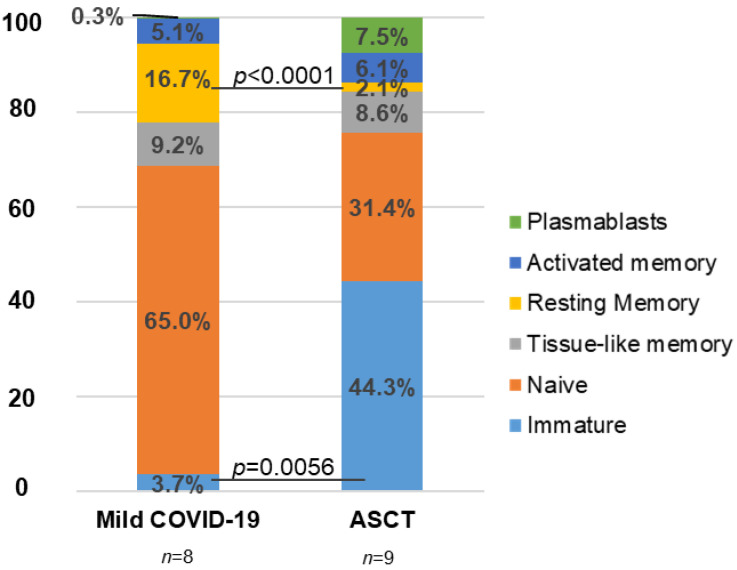
Analysis of the distribution of B cell subpopulations in individuals with oncohematological neoplasms after ASCT. Analysis by flow cytometry of B cell subpopulations in peripheral blood of patients with oncohematological disorders who underwent ASCT and healthy donors who had mild COVID-19 after staining with specific antibodies against markers CD10, CD127, CD20, and CD21. Data shown represent the mean. Statistical significance was calculated using Mann–Whitney non-parametric U test.

**Figure 2 jcm-11-02137-f002:**
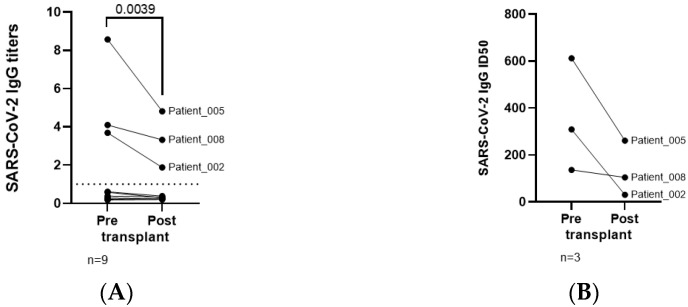
Levels of IgGs and neutralizing activity against SARS-CoV-2 in individuals with oncohematological neoplasms before and after ASCT. Analysis of anti-SARS-CoV-2 IgG levels (**A**) and anti-SARS-CoV-2 neutralizing assay represented as 50% inhibitory dose (ID50) (**B**) in plasma from patients with oncohematological disorders who underwent ASCT and healthy donors who had mild COVID-19. (**C**) Curves of neutralizing capacity in plasma from three oncohematological individuals with detectable levels of IgGs against SARS-CoV-2 before and after ASCT. Each dot represents data from one individual and lines represent mean ± the standard error of the mean (SEM). Statistical significance was calculated using Wilcoxon signed-rank test.

**Figure 3 jcm-11-02137-f003:**
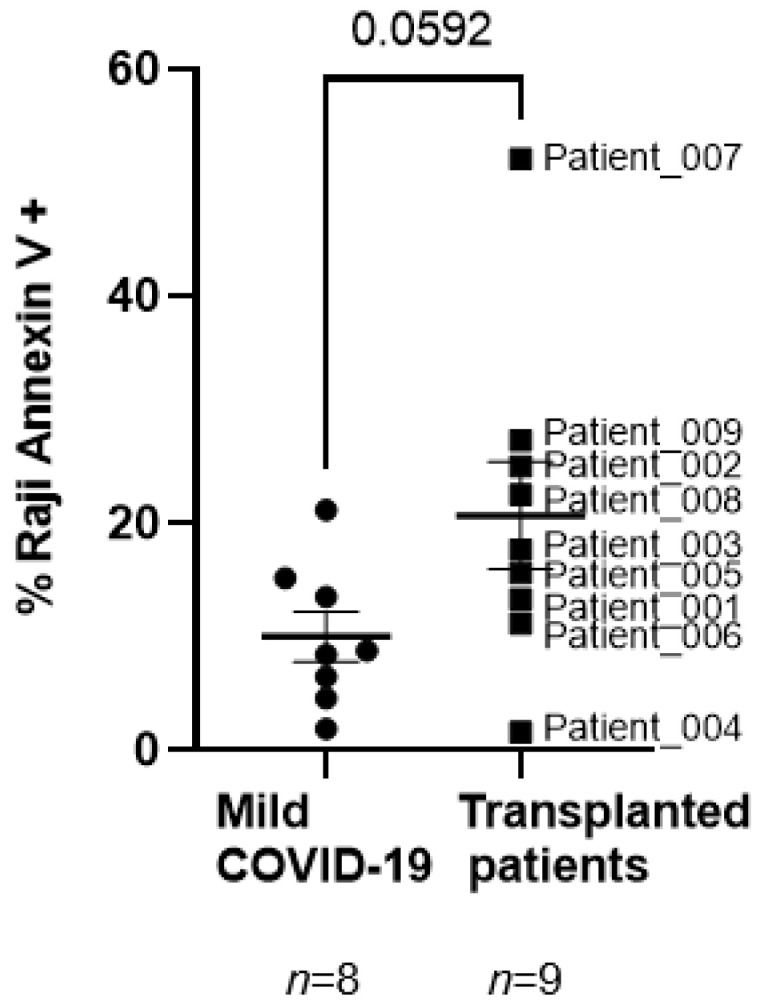
Analysis of ADCC in individuals with oncohematological neoplasms after ASCT. Quantification of early apoptosis with Annexin V-PE in PKH76-conjugated, rituximab-coated Raji cells after co-cultivation with PBMCs (1:2) from patients with oncohematological disorders who underwent ASCT and healthy donors who had mild COVID-19 as a measure of ADCC. Each dot corresponds to one sample and lines represent mean ± SEM. Statistical significance was calculated using Mann–Whitney non-parametric U test.

**Figure 4 jcm-11-02137-f004:**
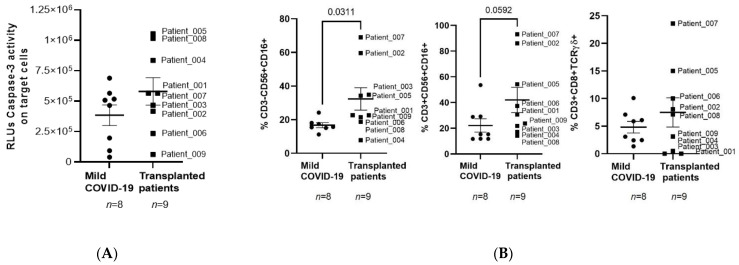
Analysis of DCC and cytotoxic cell populations in individuals with oncohematological neoplasms after ASCT. (**A**) The antiviral cytotoxicity of PBMCs from patients with oncohematological disorders who underwent ASCT and healthy donors who had mild COVID-19 was analyzed by quantifying caspase-3 activity in a monolayer of Vero E6 cells infected with pseudotyped SARS-CoV-2 viruses D614 and G614 that were co-cultured with PBMCs (1:10) for 1 h. (**B**) NK, NKT, and CD8+TCR+ cells were analyzed in supernatants of DCC assay by flow cytometry after staining with specific antibodies. Each dot corresponds to one sample and lines represent mean ± SEM. Statistical significance was calculated using the Mann–Whitney non-parametric U test.

**Figure 5 jcm-11-02137-f005:**
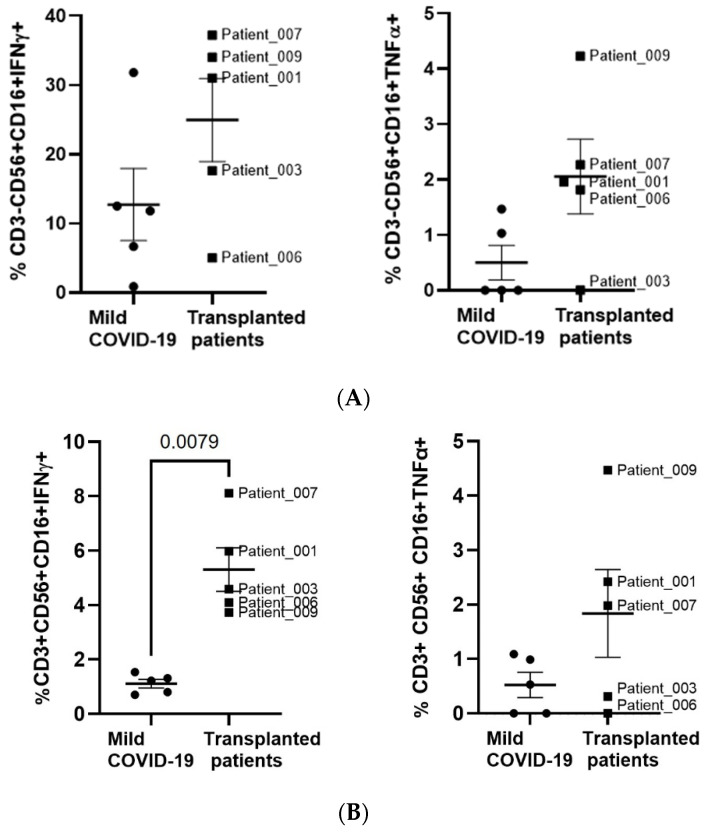
Analysis of NK, NKT and CD8+ T cell functionality in individuals with oncohematological neoplasms after ASCT. The synthesis of IFNγ and TNFα was analyzed by flow cytometry in NK (**A**), NKT (**B**), and CD8+ T cells (**C**) from patients with oncohematological disorders who underwent ASCT and healthy donors who had mild COVID-19, upon stimulation with Hsp70 or mixed SARS-CoV-2 peptides, respectively. Each dot corresponds to one sample and lines represent mean ± SEM. Statistical significance was calculated using Mann–Whitney non-parametric U test.

**Table 1 jcm-11-02137-t001:** Clinical characteristics of oncohematological patients with post-COVID-19 autologous transplantation and healthy donors with mild COVID-19 who were recruited for the study.

**Patient’s Code**	**Oncohematological Disease ***	**Gender**	**Age (Years)**	**Pre-COVID-19 Treatment ****	**Post-COVID-19 Treatment, Prior to ASCT ****	**COVID-19 Symptoms**	**Time from PCR+ to Pre-Transplant Sample (Months)**	**Time from Transplant to Post-Transplant Sample (Months)**
1	MM	Male	64	None	VRD	Asymptomatic	7.7	2.7
2	HL	Male	58	ABVD	BV-B	Asymptomatic	3.7	1.2
				BV-B				
3	MM	Male	70	None	VRD	Mild respiratory infection	9.6	1.6
4	MM	Male	71	VRD	VRD	Asymptomatic	1.6	4
5	HL	Male	35	ABVD	BV-ESHAP	Asymptomatic	4.4	6.6
6	DLBCL	Male	71	R-CHOP	PBR	Mild respiratory infection	4.7	7
7	MM	Female	64	R-ESHAP	VRD	Mild respiratory infection	5.3	6.7
				VRD				
8	MM	Male	58	None	VTD	Mild respiratory infection	3.1	1.0
9	MM	Female	56	None	VRD	Asymptomatic	9.2	1.0
**Control’s Code**	**Gender**	**Age (Years)**	**COVID-19 Symptoms**		**COVID-19 Treatment**	**Comorbidities *****	**Time from PCR+ to Sample (Months)**	
1	Female	58	Asymptomatic		None	None	2.4	
2	Female	41	Mild respiratory infection		None	None	2.8	
3	Male	29	Mild respiratory infection		None	None	2.9	
4	Female	26	Mild respiratory infection		None	None	2.9	
5	Male	62	Mild respiratory infection		HCQ	None	2.7	
6	Male	46	Mild respiratory infection		HCQ	DL	2.5	
7	Female	54	Mild respiratory infection		None	HTA	Ud	
8	Female	30	Mild respiratory infection		None	None	2.9	
9	Female	71	Mild respiratory infection		None	DL/HTA	3.2	
10	Male	70	Asymptomatic		None	HTA	Ud	

* DLBCL, diffuse large cell B lymphoma; HL, Hodgkin lymphoma; MM, multiple myeloma; ** ABVD, adriamycin-bleomycin-vinblastine-dacarbazine; BV-B, brentuximab vedotin-bendamustine; BV-ESHAP: brentuximab vedotin-etoposide-cisplatin-cytarabine-prednisone; PBR, polatuzumab vedotin-bendamustine-rituximab; R-CHOP, rituximab-cyclophosphamide-vincristine-doxorubicin-prednisone; R-ESHAP, rituximab-etoposide-cisplatin-cytarabine-prednisone; VRD, bortezomib-lenalidomide dexamethasone; VTD, bortezomib-thalidomide-dexamethasone. *** DL, dyslipidemia; HCQ, hydroxychloroquine; HTA, hypertension; Ud, undetermined.

## Data Availability

The original contributions presented in the study are included in the article. Further inquiries can be directed to the corresponding authors.

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
