# Peer review of "Strong Cellular Immune Response, but Not Humoral, against SARS-CoV-2 in Oncohematological Patients with Autologous Stem Cell Transplantation after Natural Infection"

_jcm, 2022, doi:10.3390/jcm11082137_

Round 1
Reviewer 1 Report
It is a very good article. I understand the small patient group because there are transplanted patients. The groups are homogeneous and the statistical analyses are very well done. I suggested to the authors to include more patients in order to see if the preliminary results are verified in large groups. Such analyses could give to hematological patients good information in order to be vaccinated. I suggested that the immune status acquired after vaccination plays important role in COVID-19 evolution. The article is very well written and it is very useful for our practice.
Reviewer 2 Report
In this study the Authors compared humoral and cellular immune responses of patients with multiple myeloma and lymphoma who underwent ASCT after mild or asymptomatic Sars-CoV-2 infection, to those of healthy individulas who developed similar Sars-CoV-2 infections. Despite the low number of patients included (9), this study is interesting since it reported a significant cellular responses to Sars-CoV-2 after ASCT also in patients without humoral response. The paper is well written and scientifically accurate. However, it should be noted that unlike samples from healthy individuals were collected in a range of 2.4-3.2 months after PCR positivity for Sars-CoV-2, those from patients were collected after very different times from ASCT, ranging from 1 to 7 months. This may have affected the results. Moreover, it is not clear whether patients with multiple myeloma received lenalidomide maintenance after ASCT and, also in this case, the treatment may have stimulated NK cells.
